# Fluorescence-Guided Surgery (FGS) during a Laparoscopic Redo Nissen Fundoplication: The First Case in Children

**DOI:** 10.3390/children9070947

**Published:** 2022-06-24

**Authors:** Irene Paraboschi, Laura Privitera, Stavros Loukogeorgakis, Stefano Giuliani

**Affiliations:** 1Wellcome/EPSRC Centre for Interventional & Surgical Sciences, University College London, London WC1E 6BT, UK; irene.paraboschi@policlinico.mi.it (I.P.); l.privitera@ucl.ac.uk (L.P.); 2Department of Specialist Neonatal and Paediatric Surgery, Great Ormond Street Hospital for Children NHS Foundation Trust, London WC1N 3JH, UK; stavros.loukogeorgakis@gosh.nhs.uk; 3Cancer Section, Developmental Biology and Cancer Programme, UCL Great Ormond Street Institute of Child Health, London WC1N 1EH, UK

**Keywords:** fluorescence-guided surgery, indocyanine green, Nissen fundoplication, antireflux surgery, children, real-time imaging, near-infrared, device, redo surgery

## Abstract

We present the first case of fluorescence-guided surgery (FGS) using indocyanine green (ICG) in a pediatric redo-Nissen fundoplication. The patient is a 17-year-old male with recurrent gastroesophageal symptoms who underwent primary antireflux surgery at 10 months of age. During the redo fundoplication, ICG was intravenously administered to help the visualization during the adhesiolysis between liver, stomach and right crus of the diaphragm and to spare small oesophageal vessels and the left gastric artery. In this case, FGS made the surgery easier than usual and likely reduced the risk of intra-operative complications. Therefore, we believe that this new technology should be regularly used in these types of complex intra-abdominal redo operations.

## 1. Introduction

Nissen fundoplication is one of the most common procedures performed by pediatric surgeons at resent [1,2]. Indications for anti-reflux procedures in children include gastroesophageal reflux disease (GERD) associated with failure to thrive, esophagitis, stricture formation, respiratory compromise, and aspiration pneumonia [1,2].

Even if the recent literature reports significant improvements in gastrointestinal reflux-related symptoms and a high level of patient and caregiver satisfaction following laparoscopic Nissen fundoplication, the recurrence of GERD symptoms make a redo procedure necessary in up to 2–18% of cases [2,3,4,5,6]. Therefore, a significant number of children undergo multiple procedures during their life, or ultimately require more invasive operations [7].

Laparoscopic redo surgeries are particularly demanding procedures. The instability of the camera platform, the limited motion of the laparoscopic instruments and two-dimensional (2D) imaging represent a challenge for pediatric surgeons, especially when recurrent hiatal hernias and dense postoperative adhesions significantly alter the anatomy of the esophagogastric junction [8,9,10,11]. These cases are also associated with the risk of bleeding from the liver and gastric vessels, injury to the vagus nerve, and organ perforation [8,9,10,11].

Therefore, pediatric surgeons urgently need new optical devices and technologies to reduce the intraoperative risks and facilitate the surgical procedures. The employment of near-infrared (NIR) optical imaging has been gaining popularity in many fields of pediatric surgery, where it has shown great potential to improve both surgical and oncological outcomes while minimizing anesthetic time and lowering healthcare costs [12].

To date, indocyanine green (ICG) is the most commonly adopted NIR fluorescent probe in clinical practice. ICG was developed for NIR photography by Kodak Research Laboratories in 1955 and approved for clinical use by the Food and Drug Administration (FDA) in 1956 [13]. By binding albumin, IGC is normally confined to the vascular stream and entirely excreted into the biliary tract within a few hours of its injection. Its water-solubility and fast biliary secretion make it a useful tool for hepatic and cardiac function diagnostics. Most recently, however, its clinical and surgical applications have significantly increased in both adults and children [12,14,15].

The clinical application of fluorescence-guided surgery (FGS) in laparoscopic redo Nissen fundoplication has never been reported. Therefore, we present the first step-by-step protocol to perform ICG-FGS during a laparoscopic redo Nissen fundoplication in children.

## 2. Surgical Technique

A 17-year-old male with Kabuki syndrome was referred to our tertiary center due to recurrent severe GERD symptoms, 16 years after a laparoscopic Nissen fundoplication. At the upper gastrointestinal study, the contrast was seen above the level of the left hemidiaphragm within a hiatus hernia and up to the upper esophageal level, which made a laparoscopic redo procedure necessary (Figure 1). At surgery, we encountered dense adhesions between the liver, the stomach, and the right crus of the diaphragm, which carried a high risk of bleeding. We always try not to change the gastrostomy site if it is in a good position and no gastrostomy site complications are identified. We injected ICG intravenously and use the EleVisionTM IR platform (Medtronic Ltd., Hertfordshire, UK) to visualize the main vessels of the stomach and the esophagus, to reduce the risk of bleeding during extensive adhesiolysis. This device provided high-quality images for both fluorescence and visible white light due to the automatic processing based on distance from the tissue. The operating room lights could stay on, allowing for continuous visibility of the operative field. The high sensibility and performance of the EleVisionTM IR platform (Medtronic Ltd., Hertfordshire, UK) allowed for the use of only 0.125 mg/kg (5 mg) of ICG 1 min before imaging to achieve optimal dye performance.

Compared with clinical visual intraoperative assessment, the use of ICG provided objective data that helped to identify the esophagogastric junction’s main landmarks, which were altered by the previous surgery. This facilitated the dissection through scar tissue and avoided injury to the liver parenchyma and critical blood vessels (Figure 2). Furthermore, it allowed an for objective assessment of the esophagus, the stomach perfusion during their complete mobilization and takedown of the previously failed fundoplication. The visualization of the left gastric artery (Figure 3) and the small branches of the esophageal artery (Figure 4) were excellent and clearly visible with FGS compared to the normal laparoscopic view. We were able to disconnect the esophagus and stomach from the hiatus of the diaphragm with minimal bleeding and at a good pace. We found the previous wrap to be wholly undone. We repaired the crus of the diaphragm with two stitches, and we performed a redo-Nissen fundoplication, ensuring a good segment of the intraabdominal esophagus. No postoperative complications occurred, either during hospitalization or the first 30 days postoperatively. The child was discharged home on the 5th postoperative day on full gastrostomy feeds.

## 3. Discussion

FGS has proved to be a safe tool to guide several surgical procedures in children and adults [12,14,15]. The excellent spatial resolution of fine anatomical structures, their high contrast and sensitivity, the absence of ionizing radiation, and the low cost are the main advantages of this novel intraoperative technique [12,16,17,18].

Several factors can increase the risk of intraoperative complications during a laparoscopic redo-Nissen procedure in children. The high vascularized and dense adhesions between the liver, the stomach, and the right crus of the esophagus, the challenging position of the gastrostomy, the previous sutures in the hiatus, and the partial residual wrap make this procedure particularly demanding for pediatric surgeons. Moreover, the instability of the video camera, the limited motion of the straight laparoscopic instruments, 2D imaging, and poor ergonomics for surgeons require advanced laparoscopic skills [19]. Even if most of the complications reported in the literature are minimal, we know that they can be severe and even fatal [2,3,4,5,6,8]. The most significant risks are unsuspected perforations of the esophagus or the stomach and significant intraoperative bleeding. Other potential complications include pneumothorax and injury to adjacent organs and nerves.

In his series of 118 laparoscopic redo-Nissen fundoplications, Rothenberg et al. [8]. reported that 4 patients required enterotomies during extensive lysis of adhesions, 2 patients experienced delayed gastric perforations requiring re-exploration, and 1 patient developed an incarcerated para-esophageal hernia. The two patients with gastric perforations had undergone multiple previous procedures, which led to extensive scar tissue and hiatal hernias formation. The author commented that it was likely that the stomach tissue was partially devitalized during the previous operations, resulting in weak areas that were more susceptible to injury. Conversely, in the series of redo surgeries collected by Kvello et al. [11], half of the patients experienced early postoperative complications. A third of complications were major, with two patients requiring blood transfusions due to intraoperative bleeding. Clearly, vascular injuries may be life-threatening, and FGS could have minimized the risks illustrated in these reports.

Ours is the first report of FGS in a redo laparoscopic Nissen fundoplication in children. This proved that this novel intraoperative technique is a safe and effective imaging modality in patients undergoing redo surgery, and no side effects were reported. In particular, it helped during the lysis of the dense peritoneal adhesions between the liver, the stomach, and the right crus of the diaphragm, preventing possible intraoperative bleedings and organ injuries. The use of the device significantly impacted intraoperative decision-making, helping to identify and spare the left gastric artery (Figure 3) and the esophageal vessels (Figure 4) and establishing an adequate length of the intra-abdominal esophagus. Finally, the ICG-FGS confirmed the proper vascularization of the fundus of the stomach that was used for the primary wrap and was left with reduced blood supply after ligation of short gastric vessels.

In adult general surgery, few reports examined the applications of ICG for the intraoperative evaluation of the gastroesophageal junction. In 2019, a couple of studies investigated ICG-use for the vascular perfusion of the esophageal esophagus during a laparoscopic resection of an esophageal duplication cyst [20] and the blood supply of the stapler line of the stomach during a modified sleeve gastrectomy, combined with a laparoscopic Rossetti fundoplication [21]. More recently, ICG has also been reported for the assessment of the gastric remnant’s blood supply after distal gastrectomy for an early gastric cancer [22] and for the identification of residual muscle fibers during robotic Heller-Dor procedures for esophageal achalasia [23].

The recommended dose of ICG ranges from 0.05 mg/kg to 0.5 mg/kg [12]. Following the company’s advice, we only used the dose of 0.125 mg/kg (5 mg) and obtained an excellent ICG fluorescence, without any adverse effects. The EleVisionTM IR platform (Medtronic Ltd., UK) proved to be an ultrasensitive device, which required only a small amount of ICG to perfectly delineate even the tiniest blood vessels. The device has two independent channels for visible and infrared light, making the sensitivity optimal and providing a constant near-infrared image that can be used for live surgery in parallel with normal laparoscopic vision.

When absorption spectra are used by optical-technology-based monitors, intravenously administered vital dyes, such as ICG, may result in misreading percutaneous oxygen saturation (SpO2) measured via pulse oximetry [24,25]. In patients undergoing carotid endarterectomy, lower ICG doses resulted in a smaller and less prolonged increase in regional cerebral tissue oxygen saturation (SctO2) and a lower reduction in SpO2 than higher doses [26]. Interestingly, the EleVisionTM IR platform (Medtronic Ltd., Hertfordshire, England) helped us to keep the amount of the dye low enough to not cause any anomalies in the pulse oximeter.

## 4. Conclusions

ICG fluorescence using the EleVisionTM IR platform (Medtronic Ltd., Hertfordshire, England) was safe and effective in intraoperatively evaluating the blood supply of critical vascular structures during the laparoscopic redo Nissen fundoplication. We believe that FGS will soon become part of the armoury of surgeons, significantly reducing the risk of intra- and postoperative complications.

## Figures and Tables

**Figure 1 children-09-00947-f001:**
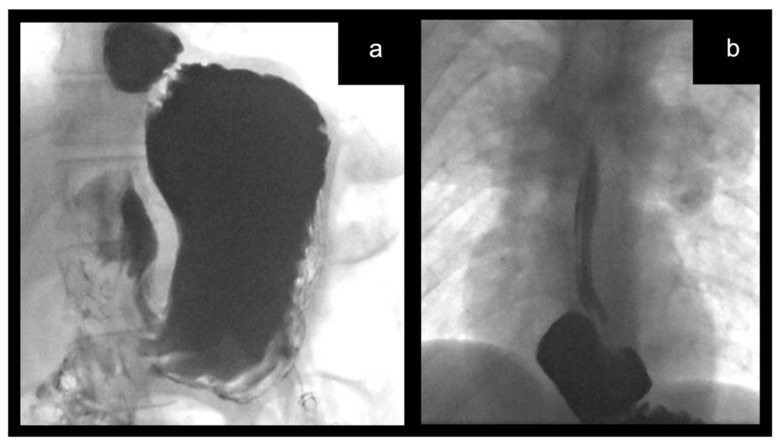
(**a**,**b**) Upper gastrointestinal contrast study showing failed fundoplication and moderate hiatal hernia with reflux to the upper esophageal level.

**Figure 2 children-09-00947-f002:**
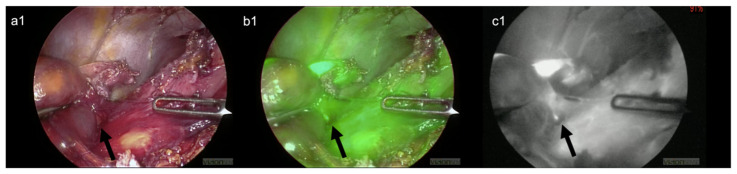
Lysis of the dense adhesions between the liver, the stomach, and the right crus of the diaphragm. EleVisionTM IR platform (Medtronic Ltd., Hertfordshire, England) allowed for the clear visualization of a vessel hidden in the tight scar tissue between liver and stomach (arrow), and this helped in the laparoscopic adhesiolysis. (**a1**) Visible-light image (**b1**) green filter image (**c1**) near-infrared (NIR) image.

**Figure 3 children-09-00947-f003:**
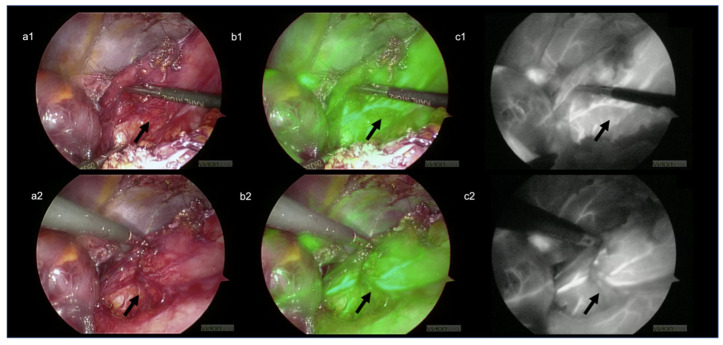
Exposure and easy visualization of the left gastric artery (arrow). (**a1**,**a2**) Visible-light image (**b1**,**b1**) green filter image (**c1**,**c2**) near-infrared (NIR) image.

**Figure 4 children-09-00947-f004:**
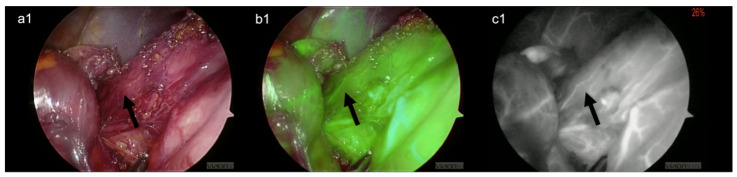
Evaluation of the remaining vascular supply of the esophagus (arrow). (**a1**) Visible-light image (**b1**) green filter image (**c1**) near-infrared (NIR) image.

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
