# Peer review of "Fluorescence-Guided Surgery (FGS) during a Laparoscopic Redo Nissen Fundoplication: The First Case in Children"

_children, 2022, doi:10.3390/children9070947_

Round 1
Reviewer 1 Report
The introductory part should be developed and more references included. It would be interesting to introduce some historical data on the introduction of ICG in surgery.
Case presentation: Some ethical considerations should be mentioned. Aspects of informed consent regarding the publication of the case are mandatory. Being a minor patient, the consent of a legal tutor is needed.
Disscussion part should be expanded. Some aspects regarding the most frequent uses of ICG are needed (biliary surgery, colorectal surgery). Cite: Serban D, Badiu DC, Davitoiu D, et al. Systematic review of the role of indocyanine green near‑infrared fluorescence in safe laparoscopic cholecystectomy (Review). Exp Ther Med 23: 187, 2022, and Alius C, Tudor C, Badiu CD, et al. Indocyanine Green-Enhanced Colorectal Surgery-between Being Superfluous and Being a Game-Changer. Diagnostics (Basel). 2020 Sep 24;10(10):742.
It is necessary to include several references.
Author Response
Dear Reviewer,
Our group would like to thank you for the opportunity to resubmit a revised version of our manuscript entitled “Fluorescence-guided surgery (FGS) during a laparoscopic redo Nissen fundoplication. The first case in children”.
Attached you find a thoroughly revised version of the manuscript with the highlighted changes from the original version. We thank you all very much for the useful comments, which have certainly improved the quality of our manuscript, and we hope it will now be acceptable for publication in “Children”.
We look forward to your comments and suggestions regarding our re-submission.
Yours sincerely.
Reviewer 2 Report
This is an excellent paper with novelty value and of great interest to pediatric surgeons. The case description is clear. I miss no keynote references. The method described should be tested by others.
Author Response
Dear Reviewer,
Our group would like to thank you for the opportunity to publish our manuscript entitled “Fluorescence-guided surgery (FGS) during a laparoscopic redo Nissen fundoplication. The first case in children” in “Children”. We hope that the method described will be tested also by others.
Yours sincerely.
Reviewer 3 Report
At the outset, I would like to congratulate the authors on a well-managed case. Several complications can occur during redo-fundoplication.
Introduction: well-written.No changes are needed.
Description of case: The case is very well described. The intraoperative images are of good quality. No changes are needed.
Discussion: I have only one comment. This is the first case description in children. However, ICG usage has been described for redo-fundoplication in the adult population in a few studies. It will be very insightful for the readers to have that one knowledge. I would like the authors to highlight the same in one paragraph.
Author Response

(The authors gave the same response as above.)

Round 2
Reviewer 1 Report
The article meets now all criteria to be published.